# Mechanical Properties and Stress–Strain Relationship of Grade 14.9 Superhigh-Tension Bolt (SHTB) Under Fire

**DOI:** 10.3390/ma18081780

**Published:** 2025-04-14

**Authors:** Xiaofang Xiao, Miao Ding, Yiqing Ge, Xiaohong Wang, Le Shen, Chunhua Ran

**Affiliations:** 1School of Architecture and Design, Chongqing College of Humanities, Science & Technology, Chongqing 401524, China; xxf3737@163.com; 2School of Civil Engineering, Chongqing University, 83 Shabeijie, Chongqing 400045, China; 3School of Materials Science and Engineering, Chongqing University, 174 Shazhengijie, Chongqing 400045, China; 20230901086g@stu.cqu.edu.cn; 4Cluster of Engineering, Singapore Institute of Technology, Singapore 138683, Singapore; miao.ding@singaporetech.edu.sg (M.D.); yiqing.ge@singaporetech.edu.sg (Y.G.)

**Keywords:** SHTB, high temperature, mechanical properties, stress–strain relationship

## Abstract

Grade 14.9 superhigh-strength bolts (SHTBs) are a type of high-strength steel bolt with a nominal tensile strength of 1400 MPa, which is significantly higher than the commonly used Grade 10.9 high-strength bolt (HSB), which has a nominal tensile strength of 1000 MPa. The use of an SHTB can reduce the number of bolts required in connections or joints, leading to material savings and improved construction efficiency. However, like HSB, the mechanical properties of an SHTB can be significantly degraded at high temperatures, though the extent of this reduction may differ. In this study, the authors designed and conducted experiments on SHTBs under elevated temperatures including both vibration and tensile coupon tests. Based on the test data, the stress–strain curves and key mechanical properties such as the Young’s modulus, yield stress, ultimate stress, ultimate strain, percentage elongation, cross-sectional area reduction, and failure strain were obtained and analyzed for various high-temperature conditions. Furthermore, a new three-stage model was proposed to describe the stress–strain relationship of SHTBs under fire conditions. Additionally, empirical formulae were developed to predict the mechanical properties of SHTBs under elevated temperatures, providing valuable insights for engineering applications and fire safety design.

## 1. Introduction

Connections are essential for transferring loads within structures and ensuring their stability. The failure of connections may undermine the structural integrity and significantly increase the risk of collapse [1,2]. The importance of robust connections is further heightened in the case of fire conditions [3]. Welding and bolt connections are commonly used connection methods for steel structures. Bolt connections can avoid the influence of the heat-affected zone [4,5,6,7] generated by welding on the performance of the joint, but the equal-strength connections may require a large number of bolts, especially for high-strength steel structures. Using Grade 14.9 super-high-strength bolts (SHTBs) can significantly reduce the number of bolts and improve the efficiency because their nominal ultimate stress is 1400 MPa, which is much higher than the 1000 MPa of the commonly used Grade 10 high-strength bolts (HSBs). SHTBs are now commercially available, enhancing the options for robust structural connections in key markets such as Japan [8,9], the United States [10,11,12], and China [13,14].

The mechanical properties of materials are pivotal for assessing the safety of structures in the event of fire hazards. Extensive research has explored the behavior of various structural steels under elevated temperatures, including S235 [15,16], S355 [17,18,19,20,21,22], S460 [23,24,25,26], S690 [27,28,29,30,31,32], and ultra-high-strength steels [33,34,35,36], each with specific temperature thresholds beyond which their properties degrade [15,16,17,18,19,20,21,22,23,24,25,26,27,28,29,30,31,32,33,34,35,36]. However, due to the different material grades and manufacturing, the mechanical properties of steel bolts are also different from structural steel. Sakumoto et al. [37] and others have shown that for F10T bolts (Grade 10.9), the ultimate stress and Young’s modulus experience rapid declines at temperatures above 400 °C. Kirby [38] reported the tensile and shear tests of Grade 8.8 bolts under high temperatures, and their results indicated that both the tensile and shear capacity of the bolts significantly decreased when the temperature exceeded 300 °C. Li et al. [39] reported the results of a series of high-temperature tensile tests of 20MnTiB (used to fabricate Grade 10.9 high-strength bolts in China). They also emphasized that the ultimate stress and Young’s modulus decreased rapidly when the temperature exceeded 300 °C. Yu et al. [40,41] conducted high-temperature double shear tests of A325 and A490 bolts (Grade 8.8 and 10.9 bolts, respectively). The results showed that the shear capacity of the A325 and A490 bolts also decreased obviously when the temperature was over 300 °C. G and Lange [42] reported a series of high-temperature tensile tests on Grade 10.9 bolts. The results also showed that the capacity was significantly less when the temperature was over 300 °C. Kodur et al. [3] conducted tests to investigate the effect of temperature on the thermal and mechanical properties of the high-strength steel used in A325 and A490 bolts. They emphasized that temperature had a significant influence on the thermal and mechanical properties of the high-strength steel used in bolts: the steel in A490 bolts possesses a slightly lower thermal conductivity than that of the A325 bolts; the yield and ultimate stress degrade at a rapid pace when the temperature exceeds 400 °C. Jonathan et al. [43] conducted high-temperature shear tests of A325 and A490 bolts, and data from these tests showed that the shear capacity of both types of bolts decreased significantly when the temperature exceeded 400 °C, but the failure displacement increased. Pang et al. [44] conducted extensive tensile experiments on the steel used for Grades 8.8, 10.9, and 12.9 bolts, respectively. Data from their tests showed that all three types of bolts exhibited a significant strength reduction when the temperature exceeded 300 °C, and when the temperature ranged from 20 to 700 °C, the Grade 12.9 bolt steel processed the maximum ultimate stress. Rezaeian et al. [45] reported the results of a series of high-temperature tensile experiments on the Grade 10.9 materials. Their study showed that the Grade 10.9 bolts experienced a rapid decrease in strength at temperatures exceeding 400 °C. Furthermore, the tension-shear combined strength [46,47], constitutive model [2], and fracture behavior [48,49] of the Grade 8.8 or 10.9 bolts (or material) were also conducted. Huseyin et al. [50] presented a study on the ductile fracture of Grade 8.8 or 10.9 bolts under fire conditions. Based on results from the tension, shear, and combined tension-shear tests, they calibrated the fracture parameters and investigated the influence of displacement at fracture values on the results by a three-bolted inclined shear test. These studies have all contributed valuable insights, but primarily focused on conventional high-strength bolts such as Grades 8.8 and 10.9. The SHTB, which has a higher ultimate stress than its conventional counterparts, has not been the subject of similar high-temperature mechanical property studies. This gap in research is notable, given the absence of data for SHTBs in the current fire resistance design guides, including BS EN 1993-1-2 [51], BS 5950-9 [52], and GB 51249-2017 [53], which limits the application of SHTBs in fire-resistant design practices.

To fill this gap, the authors designed and conducted an experimental study on the mechanical properties and stress–strain relationship of SHTBs under elevated temperatures. The Young’s modulus, yield stress, ultimate stress, ultimate strain, percentage elongation, reduction in cross-sectional area as well as failure strain of the SHTBs under various temperatures were obtained. Then, we quantitatively analyzed the variation in these mechanical properties with the increase in temperature and compared it with the public data of different types of high-strength bolts. Finally, a new three-stage model was proposed to describe the stress–strain relationship of SHTBs under fire conditions.

## 2. Experimental Investigation

### 2.1. Materials

The study utilized samples machined from Grade 14.9 SHTB, with a standard bolt size of 16 mm in diameter and 150 mm in length. The chemical composition of the SHTB, in accordance with the quality specifications from the manufacturer (Eagle Metalware Co. Ltd., Dongguan, Guangdong, China), is presented in Table 1.

### 2.2. Specimen Design

Figure 1 illustrates the dimensions and shapes of the samples utilized in this study, with Figure 1a depicting the tensile test coupon, and Figure 1b showing the vibration test samples. The tensile test coupons were designed according to GB/T 228.1-2021 [54], while the vibration test samples adhered to the guidelines of ASTM E1876-15 [55]. To assess the impact of temperature on the mechanical properties of the SHTB, we conducted a total of twenty-one tensile tests and three vibration tests.

### 2.3. Test Setup

#### 2.3.1. Vibration Tests Under Elevated Temperature

Figure 2a presents the vibration test machine (RFDA HT650, IMCE, Belgium) that was used to measure the Young’s modulus. The machine consisted of a power system, a furnace, and transducers, which convert vibration signals into electrical signals and vice versa. The high-temperature vibration test process is as follows. (1) The sample is suspended in the furnace using two copper wires, as depicted in Figure 2b. (2) The temperature is increased to the pre-selected value and held for 10 min, with a heating rate of 10 °C per minute. (3) The vibration switch is activated to measure the fundamental resonant frequency. (4) Steps 2 and 3 are repeated to measure the fundamental resonant frequency of the material at various temperature levels. Finally, the Young’s modulus is calculated from the measured fundamental resonant frequency using the formulae provided in [55] (see Equations (1) and (2)).(1)$$E_{s,T}=0.9465\left(\frac{mf_{f,T}^{2}}{b}\right)\left(\frac{L^{3}}{t^{3}}\right)T_{1}$$(2)$$T_{1}=1+6.585\left(1+0.0752\mu +0.8109\mu^{2}\right)\left(\frac{t^{2}}{L^{2}}\right)-0.868\left(\frac{t^{4}}{L^{4}}\right)-\frac{8.340\left(1+0.2023\mu +2.173\mu^{2}\right)\left(\frac{t^{4}}{L^{2}}\right)}{1+6.338\left(1+0.1408\mu +1.536\mu^{2}\right)\left(\frac{t^{4}}{L^{2}}\right)}$$
where *m* is the mass of the test specimen; *b*, *t*, and *L* are the width, thickness, and length of the test sample, respectively; $f_{f,T}$ is the fundamental resonant frequency of the specimen in flexure under a temperature of *T*; μ is the Poisson’s ratio, which was assumed as 0.3 [25]. However, the maximum operating temperature of the testing machine used in this study was 600 °C, which was the maximum temperature of the vibration tests in this study.

#### 2.3.2. Tensile Tests Under Elevated Temperature

Following [3], the steady-state test method was employed to evaluate the high-temperature mechanical properties of the SHTB. The steady-state tensile tests were conducted at nine distinct elevated temperatures: 100, 200, 300, 400, 500, 600, 650, 700, and 750 °C. To ascertain the original mechanical properties of the SHTB, three coupons were also tested without fire exposure. The heating procedure took place in an electric furnace, as depicted in Figure 3. As shown in Figure 4, two clamps were utilized to secure the test specimen to the testing machine, which was a CMT5305 universal testing machine from MTS Industrial Systems (China) Co. Ltd., Shenzhen, China.

The tensile testing procedure at elevated temperatures was as follows: (1) The test specimen was installed with only the top clamp connected to the test machine to prevent compressive loads due to thermal expansion during heating; (2) the furnace was closed and the test specimen heated to the pre-selected temperature, which was maintained for 10 min (the heating rate is 10 °C/min); (3) the bottom clamp was secured to the test machine and the tensile load applied at a displacement rate of 2 mm/minute until specimen failure [56].

## 3. Experimental Results

### 3.1. Overall Behavior

Figure 5 shows the photos of the samples after testing. All specimens failed with a ductile fracture showing a marked necking phenomenon, and the necking became more significant with temperatures larger than 500 °C. Furthermore, with an increase in the tested temperature, the surface of the tested samples gradually changed from silvery at ambient temperature (about 20 °C), to golden yellow at 300 °C, to a mix of golden-yellow and blue at 400 °C, and finally to black at 600~750 °C. Studies [57,58] have shown that oxidation on the bolt surface is the main reason for its color change. At the same time, due to the difference in oxidation products when oxidation occurs in different temperature ranges, the bolts will show different colors after experiencing different temperatures. Moreover Figure 6 presents the failure surfaces of the coupons after being tested at various high temperatures. Visible necking phenomenon and shear lips could be seen from the photos, which became more significant with the increase in temperature, also indicating that the SHTB possessed ductile failure characteristics [2,13].

### 3.2. Stress–Strain Curves

The typical engineering stress–strain curves of the SHTB at various temperatures are summarized in Figure 7. The stress–strain curves exhibited significant elastic and plastic behaviors, and no yield plateau was observed. Furthermore, the coupons were exposed to a temperature equal to or below 400 °C, the shape of the engineering stress–strain curves of coupons were similar, and the engineering strain at fracture changed little. However, when the coupons underwent a temperature exceeding 500 °C, the ultimate stress mark decreased with the increasing temperature, while a reverse trend was observed on the engineering strain. In addition, the yield stress (0.2% proof stress), ultimate stress, and the corresponding strain (ultimate strain) were also obtained from these engineering strain–stress curves.

### 3.3. Mechanical Properties

Table 2 summarizes the mechanical properties of the SHTB under various temperatures (the average of coupons in each group) including the Young’s modulus ($E_{s,T}$), proportional limit ($f_{p,T}$), yield stress ($f_{y,T}$), ultimate stress ($f_{u,T}$), ultimate strain ($\varepsilon_{u,T}$), the percentage elongation after fracture ($A_{T}$), the percentage reduction in area ($Z_{T})$, failure strain ($\varepsilon_{f,T}$), and corresponding stress ($f_{f,T}$). The percentage reduction in area, an index presenting the ductility of steel, was calculated as the ratio of the maximum reduction in the cross-section area from the initial area of the specimen [59]; the failure strain was calculated using Equation (3) [60], where $r_{T}$ is the radius of the minimum cross-section of the coupon tested at a temperature of T, and $r_{0,T}$ is the initial value of $r_{T}$.(3)$$\varepsilon_{f,T}=2ln(\frac{r_{T}}{r_{0,T}})$$

#### 3.3.1. Young’s Modulus

The deterioration of the mechanical properties of the SHTB under elevated temperature is represented by reduction factors. For the Young’s modulus, the reduction factor was defined as the ratio of the Young’s modulus from a specific temperature ($E_{s,T}$) to that of the coupons at ambient temperature ($E_{s,Ambient}$). Figure 8a presents the Young’s modulus reduction factor of SHTB in this test, along with that of conventional high-strength bolts (Grades 8.8, 10.9, and 12.9) in the previous literature [2,37,39,44,45,48,49]. It can be found that the Young’s modulus reduction factor of the SHTB decreased with increasing temperature; a similar trend can also be found in the public data. However, when the temperature exceeded 400 °C, the Young’s modulus reduction factor of SHTB was larger than that of the conventional high-strength bolts. Figure 8b illustrates the comparisons of the Young’s modulus reduction factor of the SHTB with the predicted formulae suggested by previous studies. It was found that these existing formulae could not well predict the Young’s modulus reduction factor of the SHTB because these formulae were proposed based on the test data of conventional high-strength bolts. According to Wang et al. [25], an exponent formula was proposed to predict the Young’s modulus reduction factor of the SHTB, as shown in Equation (4). As shown in Figure 8b, the proposed formula could well predict the test data in this study.(4)$$\frac{E_{s,T}}{E_{s,Ambient}}=1.113-0.111\exp\left(\frac{T-20}{566}\right)\;20\;{}^{\circ}C<T\leq 750\;{}^{\circ}C$$

#### 3.3.2. Proportional Limit

The proportional limit reduction factor was defined as the ratio of the proportional limit under a specific temperature level ($f_{p,T}$) to the yield stress of the coupons at ambient temperature ($f_{y,Ambient}$). Figure 9 presents the comparisons of the proportional reduction factor from this study and EC3. As shown in Figure 9, there was an obvious difference between the proportional limit reduction factors from this study and EC3. When the temperature was not higher than 100 °C, the proportional limit reduction factors from EC3 were one, significantly larger than the corresponding value of the SHTB; however, both of them decreased with the increase in temperature. However, when the temperature was not lower than 200 °C, the proportional limit reduction factor of the SHTB was larger than that from EC3, but the difference decreased with the increase in temperature. This may have been caused by the difference in delivery conditions of the structural steel and SHTB. Based on the experimental results, the authors’ recommended predictive formulae for the proportional limit reduction factor of the SHTB under various temperatures are proposed, as shown in Equation (5).(5)$$\frac{f_{p,T}}{f_{y,Ambient}}=\left\{\begin{array}{l} 0.900\;\;\;\;\;\;\;\;\;\;\;\;\;\;\;\;\;\;\;\;\;\;\;\;\;\;\;\;\;\;\;\;\;20\;{}^{\circ}C\leq T\leq 238\;{}^{\circ}C\\ 0.900-0.252\frac{T-238}{288}\;\;238\;{}^{\circ}C<T\leq 526\;{}^{\circ}C\\ 0.057+0.603\frac{750-T}{224}\;\;526\;{}^{\circ}C<T\leq 750\;{}^{\circ}C \end{array}\right.$$

#### 3.3.3. Yield Stress

The yield stress reduction factor was defined as the ratio of the yield stress under elevated temperature level ($f_{y,T}$) to that of the coupons at ambient temperature ($f_{y,Ambient}$). Figure 10 presents the comparisons of the yield stress, yield stress reduction factor, and prediction formulae from this study and other public studies. As shown in Figure 10a, it can be found that the SHTB exhibited a much larger yield stress than that of the conventional high-strength bolts under the same temperature. Figure 10b presents the yield stress reduction factor of the SHTB in this test and that of conventional high-strength bolts in previous studies [2,3,37,39,44,45,48,49]. It can be found that the yield stress reduction factor of SHTB was much larger than that of the Grade 12.9 high-strength bolt and close to the upper limit of the data from the Grades 8.8 and 10.9 high-strength bolts. Moreover, the authors also compared the prediction formulae of the yield stress reduction factor from previous studies, as shown in Figure 10c. It can be found that when the temperature was at a level equal to or below 400 °C, the yield stress reduction factor of the Grade 14.9 SHTB was close to the prediction made by the formula suggested by Kodur et al. [3], where the reduction factor maintained a constant of one. However, when the temperature level exceeded 400 °C, the yield stress reduction factor was slightly larger than the prediction made by the formula suggested by Ban et al. [2], which is based on Grade 10.9 high-performance (HP) bolts. Based on the experimental results, the authors’ recommended predictive formulae for the yield stress reduction factor of the SHTB under various temperatures are proposed, as shown in Equation (6).(6)$$\frac{f_{y,T}}{f_{y,Ambient}}=\frac{2.522}{1+{\left(T/800\right)}^{3.4}}-1.342\leq 1\;\;\;\;\;\;\;\;20\;{}^{\circ}C<T\leq 750\;{}^{\circ}C$$

#### 3.3.4. Ultimate Stress

The ultimate stress reduction factor was defined as the ratio of the ultimate stress under a specific temperature level ($f_{u,T}$) to that of the coupons at ambient temperature ($f_{u,Ambient}$). Figure 11 presents the comparisons of the ultimate stress, ultimate stress reduction factor, and prediction formulae from this study and other public studies. Similar to the comparisons of the yield stress, the SHTB also exhibited a much larger ultimate stress than that of the conventional high-strength bolts under the same temperature, as shown in Figure 11a. This means that the SHTB can provide a larger capacity than the high-strength bolts when affected by the same fire hazard. Figure 11b presents the ultimate stress reduction factor of the SHTB in this study and that of the conventional high-strength bolts in previous studies [2,3,37,39,44,45,48,49]. It was found that the ultimate stress reduction factor of the SHTB was also much larger than that of the Grade 12.9 high-strength bolt when the temperature exceeded 200 °C, which indicates a marked difference from that of the Grades 8.8 and 10.9 high-strength bolts. In addition, the authors also compared the prediction formulae of the yield stress reduction factor from previous studies, as shown in Figure 11c. It was found that Equation (6) could also well predict the ultimate stress reduction factor of the SHTB.

#### 3.3.5. Ultimate Strain

The ultimate strain reduction factor was defined as the ratio of the ultimate strain under a specific temperature level ($\varepsilon_{u,T}$) to that of the coupons at ambient temperature ($\varepsilon_{u,Ambient}$). Figure 12 presents the comparisons of the ultimate strain, ultimate strain reduction factor, and prediction formulae from this study and other public studies. As shown in Figure 12a, it was found that the ultimate strain of the SHTB was much smaller than that of the high-strength bolts with the same temperature. However, with the increase in temperature, the ultimate strain of the SHTB was steadier than that of the high-strength bolts (exhibited a smaller change). Given the smaller ultimate strain, the ultimate strain reduction factor of the SHTB ranged from the lower and upper results of the data for high-strength bolts, as shown in Figure 12b. As shown in Figure 12c, the existing formulae in previous studies could not well predict the ultimate strain reduction factor of the SHTB. Thus, the use of Equation (7) as the prediction formula for the ultimate strain reduction factor of the SHTB was recommended. This formula is in good agreement with the experimental data.(7)$$\frac{\varepsilon_{u,T}}{\varepsilon_{u,Ambient}}=\left\{\begin{array}{l} 3.372-8.81\frac{T}{1000}-6.75{\left(\frac{T}{1000}\right)}^{2}\leq 1\;\;\;\;20\;{}^{\circ}C<T\leq 700\;{}^{\circ}C\\ 0.561+1.84\frac{T-700}{100}\;\;\;\;\;\;\;\;\;\;\;\;\;\;\;\;\;\;\;\;\;\;\;\;\;\;\;\;\;\;\;\;\;\;700\;{}^{\circ}C<T\leq 750\;{}^{\circ}C \end{array}\right.$$

#### 3.3.6. Reduction in Cross-Section Area

The reduction factor of reduction in cross-section area (reduction factor of reduction in area) is defined as the ratio of the reduction in cross-section area (reduction in area) under a specific temperature level ($Z_{T}$) to that of the coupons at ambient temperature ($Z_{Ambient}$). As shown in Figure 13a, the reduction in area of the SHTB exhibited an increasing trend with the increasing temperature and was much smaller than that of the high-strength bolts when the temperature exceeded 600 °C, but the opposite was true when the temperature exceeded 600 °C. Because of this reason, the reduction factor of the reduction area of the SHTB was much higher than that of the high-strength bolts when the temperature was higher than 600 °C, as shown in Figure 13b. Therefore, Equation (8) is recommended as the prediction formula for the reduction factor of the percentage reduction in area of the SHTB. This formula is in good agreement with the experimental data.(8)$$\frac{Z_{T}}{Z_{Ambient}}=\left\{\begin{array}{l} 1\;\;\;\;\;\;\;\;\;\;\;\;\;\;\;\;\;\;\;\;\;\;\;\;\;\;\;\;\;\;\;\;\;\;\;\;20\;{}^{\circ}C<\;T\leq 300\;{}^{\circ}C\\ 1+1.078\frac{T-300}{750-300}\;\;300\;{}^{\circ}C<T\leq 750\;{}^{\circ}C \end{array}\right.$$

#### 3.3.7. Failure Strain and Stress

Figure 14 presents the reduction factors of the failure strain and stress of the SHTB. The reduction factor of failure strain was defined as the ratio of the failure strain under a specific temperature level ($\varepsilon_{f,T}$) to that of the coupons at ambient temperature ($\varepsilon_{f,Ambient}$), while the reduction factor of failure stress was defined as the ratio of the failure stress under a specific temperature level ($f_{f,T}$) to the yield stress under ambient temperature ($f_{y,Ambient}$).

From Figure 14a, it can be found that: (1) when the temperature does not exceed 300 °C, the reduction factor of failure strain almost maintained a constant of one; and (2) when the temperature exceeded 300 °C, the reduction factor of failure strain significantly increased with the increase in temperature. Equation (9) was developed to predict the reduction factor of the failure strain of the SHTB, which provided good predictions of the test data. However, the reduction factor of failure stress showed a different trend to the reduction factor of failure strain: the value of the reduction factor of failure stress first increased and then decreased as the temperature increased and reached the maximum value at 300 °C; when the temperature was not lower than 700 °C, the value of the reduction factor of failure stress was almost zero. Based on the fitting analysis, Equation (10) was suggested to predict the reduction factor of the failure stress of the SHTB, as shown in Figure 14b.(9)$$\frac{\varepsilon_{f,T}}{\varepsilon_{f,Ambient}}=\left\{\begin{array}{l} 1+0.542\frac{T-20}{1000}\;\;\;\;\;\;\;\;\;\;\;\;\;\;\;20\;{}^{\circ}C\leq T\leq 320\;{}^{\circ}C\\ 1.016+5.549\frac{T-320}{1000}\;\;\;\;300\;{}^{\circ}C<T\leq 660\;{}^{\circ}C\\ 2.903+74.58\frac{T-660}{1000}\;\;\;660\;{}^{\circ}C<T\leq 750\;{}^{\circ}C \end{array}\right.$$(10)$$\frac{f_{f,T}}{f_{y,Ambient}}=\left\{\begin{array}{l} 0.763+0.133\frac{T-20}{280}-0.060{\left(\frac{T-20}{280}\right)}^{2}\;\;\;\;\;\;\;\;20\;{}^{\circ}C\leq T\leq 300\;{}^{\circ}C\\ 0.836-1.029\frac{T-300}{400}+0.193{\left(\frac{T-300}{400}\right)}^{2}\;300\;{}^{\circ}C<T\leq 750\;{}^{\circ}C \end{array}\right.$$

## 4. Stress–Strain Relationship

Uniaxial stress–strain models are commonly used in the component model for structural steel joints [60]. However, to evaluate the mechanical behavior of joints from elastic to failure, a stress–strain model with a descending stage is necessary. Zhu et al. [60] suggested a trilinear model, which can be determined with the yield point, peak point, and failure point, as shown in Equation (11).(11)$$\sigma =\left\{\begin{array}{l} E_{s,T}\varepsilon \;\;\;\;\;\;\;\;\;\;\;\;\;\;\;\;\;\;\;\;\;\;\;\;\;\;\;\;\;\;\;\;\;\;\;\;\;\;\;\;\;\;\;\;\;\;\;\;\;\;\;\;\varepsilon \leq f_{y,T}/E_{s,T}\\ f_{y,T}+\left(f_{u,T}-f_{y,T}\right)\frac{\varepsilon -f_{y,T}/E_{s,T}}{\varepsilon_{u,T}-f_{y,T}/E_{s,T}}\;\;\;\;\;\;\varepsilon \leq \varepsilon_{p,T}\\ f_{u,T}+\left(f_{f,T}-f_{u,T}\right)\frac{\varepsilon -\varepsilon_{u,T}}{\varepsilon_{f,T}-\varepsilon_{u,T}}\;\;\;\;\;\;\;\;\;\;\;\;\;\;\;\;\varepsilon \leq \varepsilon_{f,T} \end{array}\right.$$

However, the trilinear model is so simplified that it may not accurately describe the stress–strain curve of the SHTB. Therefore, the authors proposed a new three-stage stress–strain model (see Equation (12)): (1) elastic stage (before the proportional limit), in this stage the strain–stress curve is linear; (2) the strain hardening stage (from the proportional limit to the peak stress), in this stage, the curve is assumed to be an exponential curve, where the parameter $m_{T}$ can be calculated by the strain and stress of the yield point (with a plastic strain of 0.2%); and (3) the descending stage (after peak stress), where the curve is assumed as an arc and the parameter $a_{T}$ controls the radius of curvature of the curve. Based on the analytical results, the values of parameters $m_{T}$ and $a_{T}$ of the SHTB with different temperatures are summarized in Table 3.(12)$$\sigma =\left\{\begin{array}{l} E_{s,T}\varepsilon \;\;\;\;\;\;\;\;\;\;\;\;\;\;\;\;\;\;\;\;\;\;\;\;\;\;\;\;\;\;\;\;\;\;\;\;\;\;\;\;\;\;\;\;\;\;\;\;\;\;\;\;\;\;\;\;\;\;\;\;\;\;\;\;\;\;\;\;\;\;\;\;\;\;\;\;\;\;\;\;\;\;\;\;\;\;\;\;\;\;\;\;\;\;\;\;\;\;\;\;\varepsilon \leq \varepsilon_{p,T}=f_{p,T}/E_{s,T}\\ f_{u,T}-\left(f_{u,T}-f_{p,T}\right)\left[\exp\left(-m_{T}\frac{\varepsilon -\varepsilon_{p,T}}{\varepsilon_{u,T}-\varepsilon_{p,T}}\right)-\frac{\varepsilon -\varepsilon_{p,T}}{\varepsilon_{u,T}-\varepsilon_{p,T}}\exp\left(-m_{T}\right)\right]\;\;\;\varepsilon \leq \varepsilon_{p,T}\\ f_{f,T}+\left(f_{u,T}-f_{f,T}\right)\left(\frac{1}{2}-\frac{\sqrt{2}}{2}b_{T}+\sqrt{b_{T}^{2}-{\left(\frac{\varepsilon -\varepsilon_{u,T}}{\varepsilon_{f,T}-\varepsilon_{u,T}}-\frac{1}{2}+\frac{\sqrt{2}}{2}c_{T}\right)}^{2}}\right)\;\varepsilon \leq \varepsilon_{f,T} \end{array}\right.$$(13)$$\left\{\begin{array}{l} b_{T}=\frac{1}{4a}-\frac{a_{T}}{2}\\ c_{T}=b_{T}+a_{T} \end{array}\right.$$

Figure 15 presents the comparisons of the stress–strain curves of the SHTB from the tests, the tri-linear model, and the proposed model. It was found that both the trilinear model and the proposed model described the characteristics of stress–strain curves of the SHTB under various temperatures. However, the curves from the proposed model agreed more with the test data and also showed a difference of proportional limit and proved yield stress at 0.2% plastic strain. Therefore, the proposed model is recommended for component models for structural steel joints, and the trilinear model is more recommended for simplified analysis.

## 5. Conclusions

In this study, a series of vibration tests and tensile tests of Grade 14.9 SHTB coupons were performed at elevated temperatures. The stress–strain curves, Young’s modulus, yield stress, ultimate stress, ultimate strain, percentage elongation, reduction in the cross-section area as well as failure strain were obtained and discussed. Based on the obtained results, the following conclusions can be drawn:(1)Similar to the conventional high-strength bolts (Grades 8.8, 10.9, and 12.9), the surface states of the SHTB changed to some extent after the test. As the temperature increased, the surface color changed from silvery to black, and the “blue brittle” phenomenon was observed when the temperature was around 400 °C.(2)When the temperature was below 400 °C, the deterioration in the Young’s modulus of the SHTB was similar to the conventional high-strength bolts, but the SHTB had a larger value when the temperature exceeded 400 °C.(3)Both the yield stress and ultimate stress of the SHTB were much larger than that of the conventional high-strength bolts under the same temperature. When the temperature did not exceed 400 °C, there were few changes in the yield stress and ultimate stress but exhibited a marked decreasing trend when the temperature exceeded 400 °C.(4)Both the reduction in the cross-section area and failure strain of the SHTB exhibited an increasing trend with the increase in temperature. However, the failure stress first increased with the temperature increase, and then decreased with the increase in temperature.(5)Simplified prediction formulae for the reduction factors of the Young’s modulus, proportional limit, yield stress, ultimate tensile stress and strain, failure stress and strain, and the reduction in the cross-section area of the SHTB are suggested. These formulae could well predict the reduction factor of the mechanical properties of the SHTB with the elevated temperature varying from 20 to 750 °C. These formulae can be used to perform future fire safety designs and evaluation.(6)A three-stage model is suggested to describe the full uniaxial stress–strain curves of the SHTB under various elevated temperatures.

## Figures and Tables

**Figure 1 materials-18-01780-f001:**
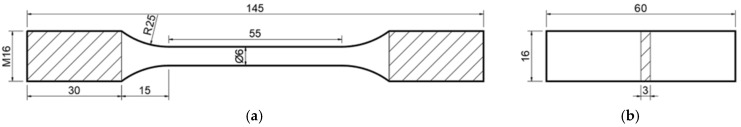
Dimensions of the test samples (unit: mm). (**a**) Tensile test sample; (**b**) vibration test sample.

**Figure 2 materials-18-01780-f002:**
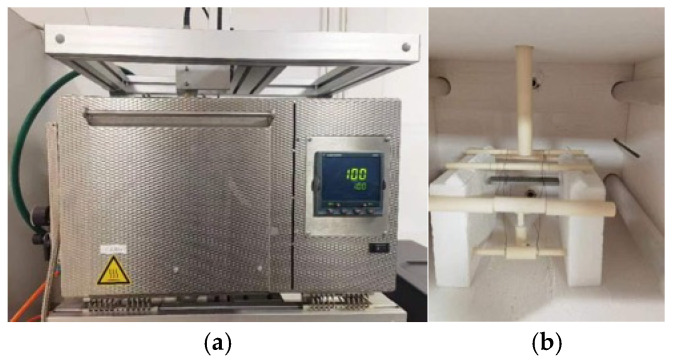
Test machine and specimen setup: (**a**) vibration test machine; (**b**) setup.

**Figure 3 materials-18-01780-f003:**
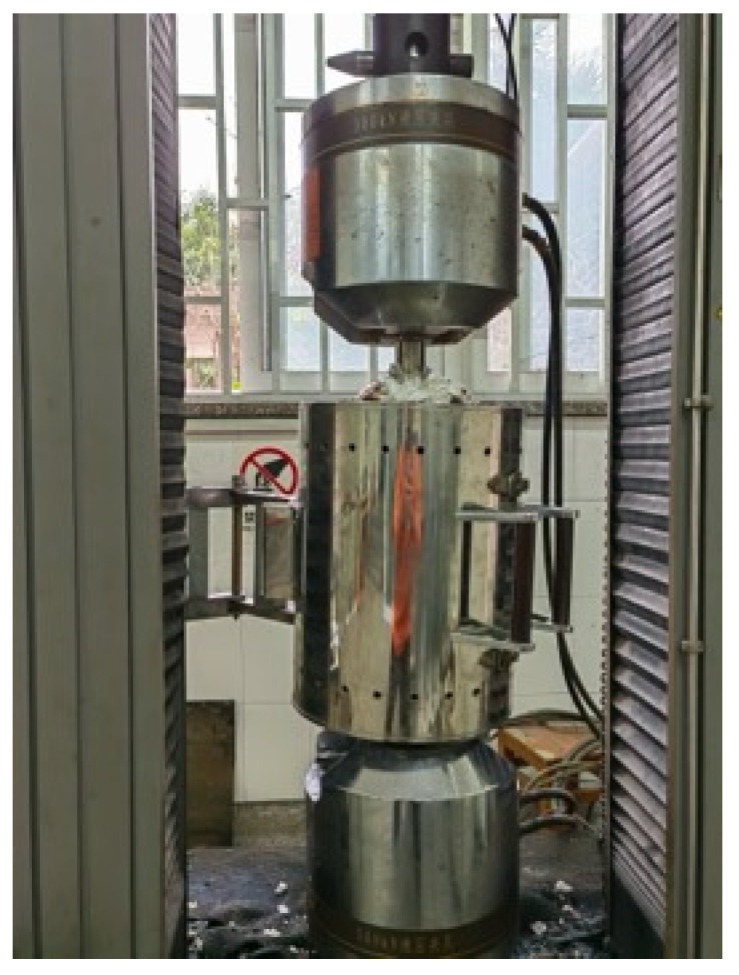
Test equipment.

**Figure 4 materials-18-01780-f004:**
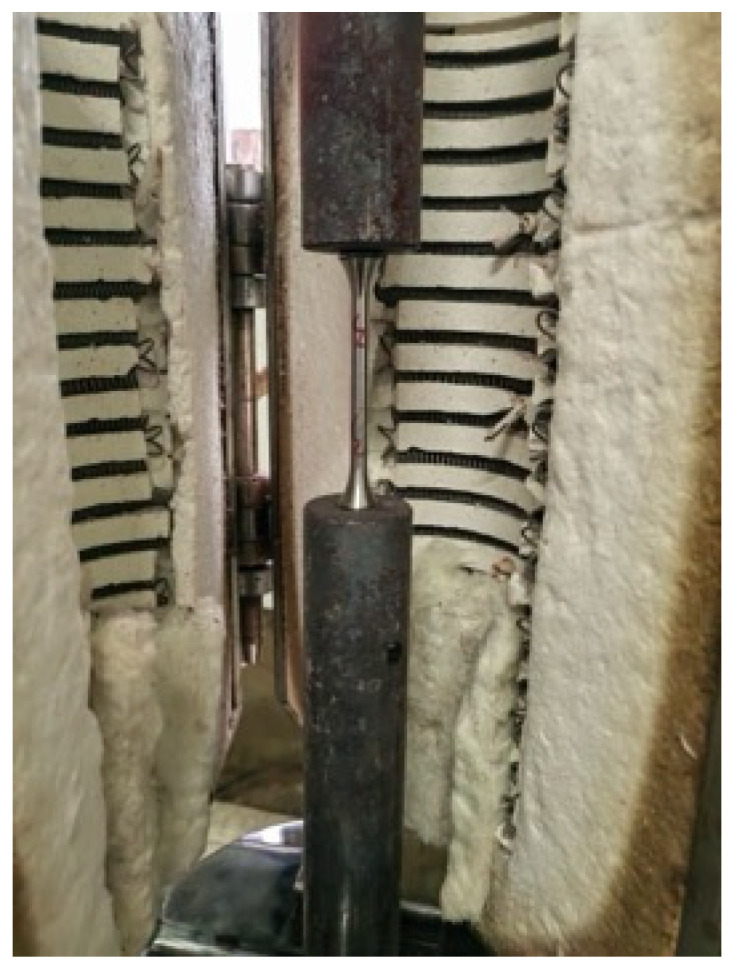
Installed specimen and the clamps.

**Figure 5 materials-18-01780-f005:**
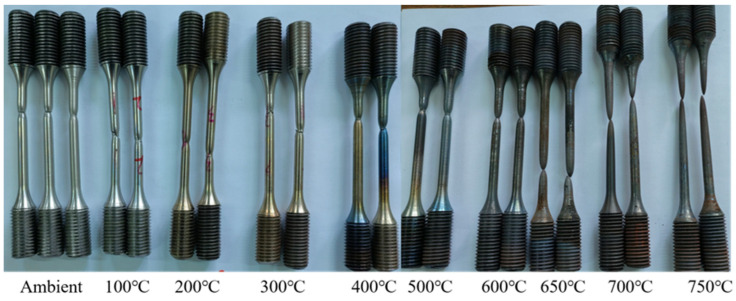
The test samples after testing.

**Figure 6 materials-18-01780-f006:**
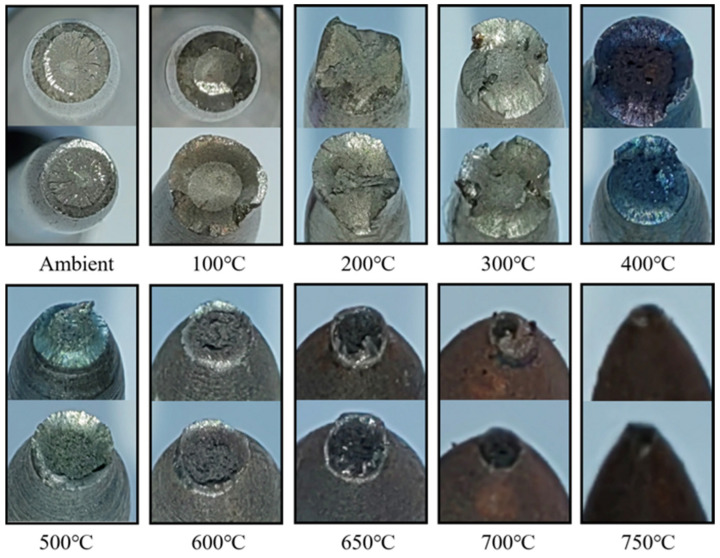
Fracture surfaces of the coupons under various temperatures.

**Figure 7 materials-18-01780-f007:**
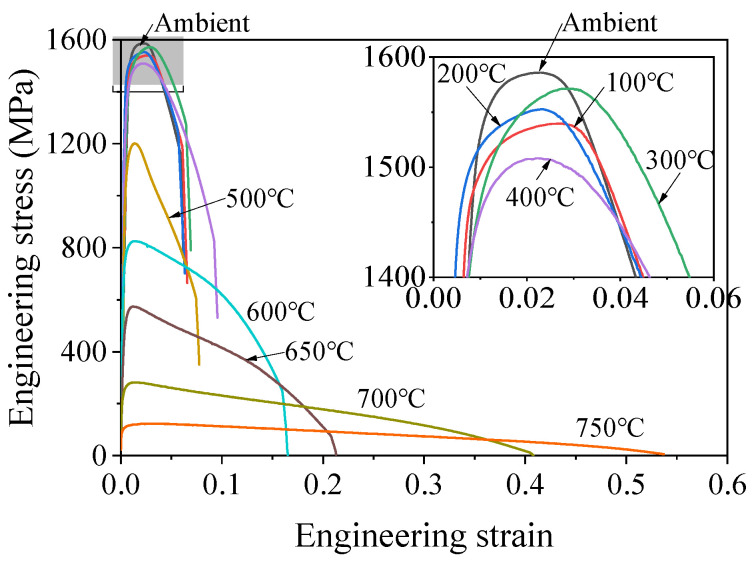
Typical stress–strain curves of an SHTB under various temperatures.

**Figure 8 materials-18-01780-f008:**
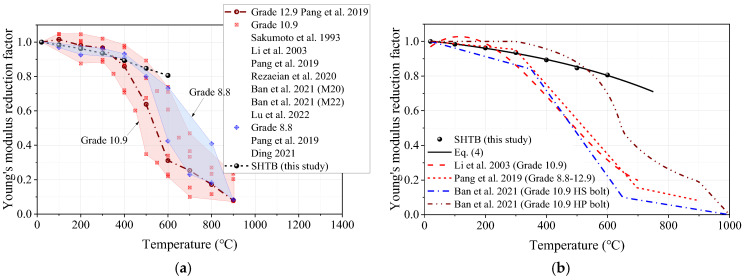
The Young’s modulus reduction factors. (**a**) Comparison of the public data; (**b**) comparison of the formulae in the previous study [2,37,39,44,45,48,49].

**Figure 9 materials-18-01780-f009:**
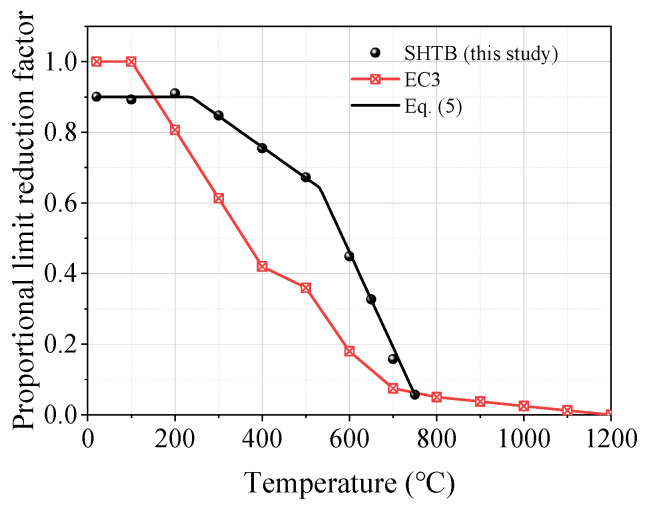
Proportional limit and the corresponding reduction factors.

**Figure 10 materials-18-01780-f010:**
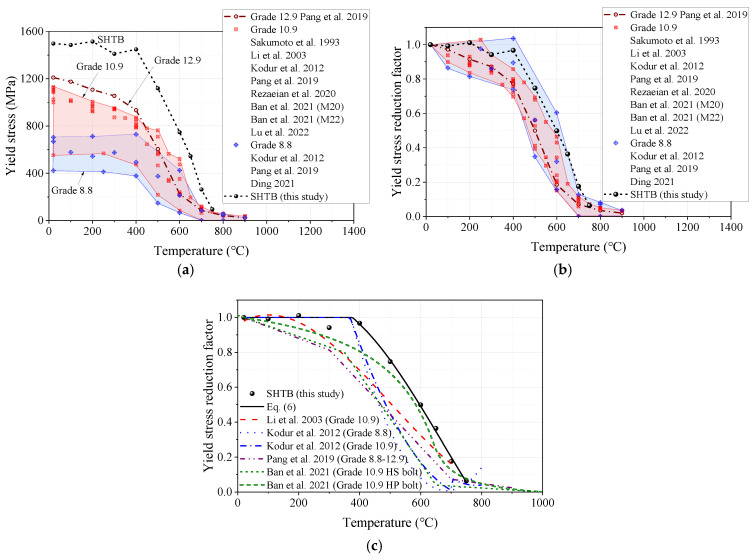
Yield stress and the corresponding reduction factors: (**a**) yield stress; (**b**) yield stress reduction factors; (**c**) comparison of different predicted formulae [2,3,37,39,44,45,48,49].

**Figure 11 materials-18-01780-f011:**
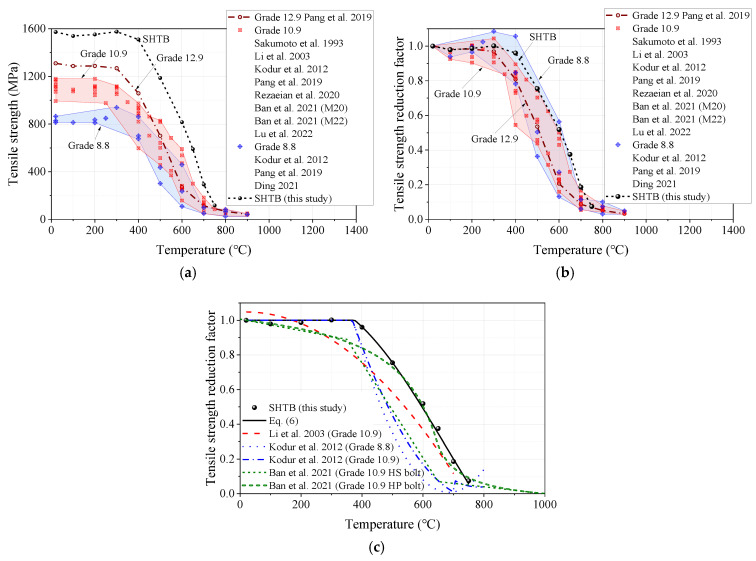
Ultimate stress and the corresponding reduction factors: (**a**) ultimate stress; (**b**) ultimate stress reduction factors; (**c**) comparison of different predicted formulae [2,3,37,39,44,45,48,49].

**Figure 12 materials-18-01780-f012:**
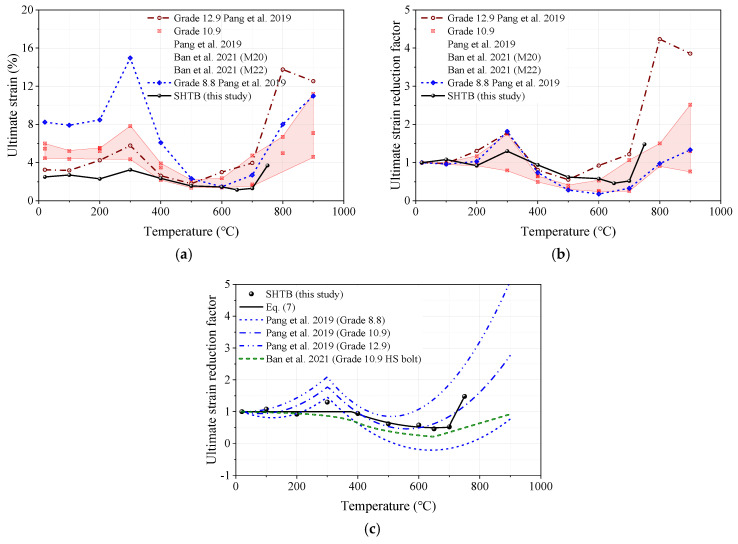
Ultimate strain and the corresponding reduction factors: (**a**) ultimate strain; (**b**) ultimate strain reduction factors; (**c**) comparison of different predicted formulae [2,44].

**Figure 13 materials-18-01780-f013:**
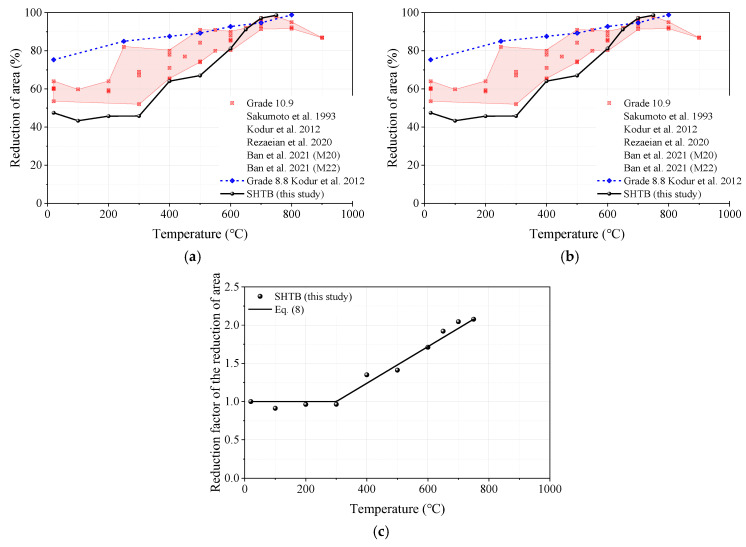
Reduction factors of reduction in the cross-section area: (**a**) reduction in the cross-section area; (**b**) reduction factors of reduction in the cross-section area; (**c**) prediction formula [2,3,37,45].

**Figure 14 materials-18-01780-f014:**
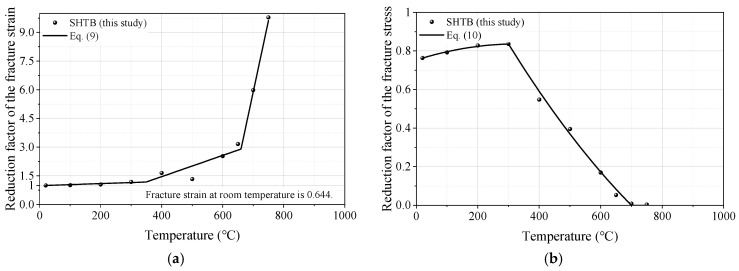
Prediction formula of the reduction factors of failure strain and stress: (**a**) failure strain; (**b**) failure stress.

**Figure 15 materials-18-01780-f015:**
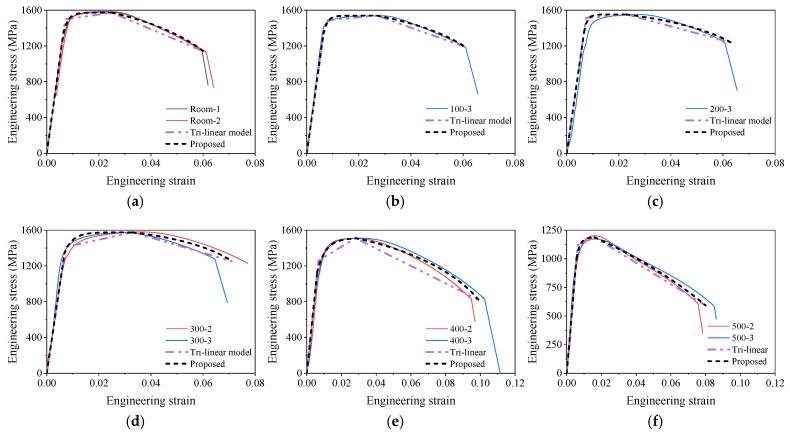
Comparisons of the stress–strain curves from test and proposed model: (**a**) room temperature; (**b**) 100 °C; (**c**) 200 °C; (**d**) 300 °C; (**e**) 400 °C; (**f**) 500 °C; (**g**) 600 °C; (**h**) 650 °C; (**i**) 700 °C; (**j**) 750 °C.

**Table 1 materials-18-01780-t001:** Chemical composition of the SHTB (%).

C	Si	Mn	P	S	Mo	Cr	Ni	Cu	V
0.4	0.19	0.51	0.015	0.012	0.57	0.99	0.11	0.09	0.28

**Table 2 materials-18-01780-t002:** Summary of the mechanical properties of the SHTB under elevated temperature.

T °C	$E_{s,T}$ GPa	$f_{p,T}$ MPa	$f_{y,T}$ MPa	$\varepsilon_{u,T}$	$f_{u,T}$ MPa	$Z_{T}$ %	$\varepsilon_{f,T}$	$f_{f,T}$ MPa
20	211.7	1348.8	1497.9	0.025	1573.3	47.5	0.060	1143.4
100	208.2	1337.2	1485.8	0.027	1539.6	43.3	0.061	1187.1
200	203.7	1363.3	1514.8	0.023	1552.6	45.7	0.063	1240.5
300	197.9	1269.5	1410.6	0.033	1575.7	45.8	0.071	1253.0
400	189.1	1130.6	1256.2	0.028	1510.2	64.1	0.099	820.8
500	179.3	1007.4	1118.9	0.016	1187.1	67.0	0.080	592.2
600	170.6	672.2	746.8	0.015	816.0	81.2	0.152	254.7
650	164.1 ^1^	489.9	544.6	0.012	590.4	91.2	0.190	80.8
700	157.5 ^1^	236.5	262.9	0.013	292.9	97.1	0.359	12.4

^1^ Data were from Section 3.3.1.

**Table 3 materials-18-01780-t003:** Summary of the values of parameters $m_{T}$ and $\alpha_{T}$ for the SHTB under various temperatures.

T °C	$m_{T}$	$\alpha_{T}$	T °C	$m_{T}$	$\alpha_{T}$
20	10.162	0.106	500	5.011	0.054
100	13.632	0.106	600	4.007	0.160
200	13.136	0.123	650	3.475	0.103
300	8.210	0.142	700	3.577	0.076
400	4.406	0.141	750	6.611	0.097

## Data Availability

The original contributions presented in this study are included in the article. Further inquiries can be directed to the corresponding authors.

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
