# Peer review of "Mechanical Properties and Stress–Strain Relationship of Grade 14.9 Superhigh-Tension Bolt (SHTB) Under Fire"

_materials, 2025, doi:10.3390/ma18081780_

Round 1
Reviewer 1 Report
Comments and Suggestions for Authors
This manuscript is good study about the temperature effect on the mechanical properties of the SHTB. The texting were correctly carried out and the date were employed to develop a model. The surface analyses of the samples after testing was adequate and the results were compared with literature information. For these reasons, the manuscript can be published in the current form. However, the next comments could be considered.
The suppliers of the bolts, furnace and texting machine should be added in the 2. Experimental investigation because the readers can be interesting to know a good suppler.
An explain about the various colour found the samples should be discussed in the manuscript because this is curious results, this can be due to the different oxidation grade.
My specific comments are the following;
To add reference in line 160 of page 5 "istics [REF]. "
To insert reference in line 283 of page 10 "well predict the ultimate strain reduction factor of SHTB. Thus, the authors [REF] recommended"
To include reference in line 300 of the page 11 "higher than 600℃, as shown in Figure 13(b). Therefore, the authors [REF] recommended predic-"
Author Response
This manuscript is good study about the temperature effect on the mechanical properties of the SHTB. The texting were correctly carried out and the date were employed to develop a model. The surface analyses of the samples after testing was adequate and the results were compared with literature information. For these reasons, the manuscript can be published in the current form. However, the next comments could be considered.
Reply: Thanks for your positive comment.
Comment 1: The suppliers of the bolts, furnace and testing machine should be added in the 2. Experimental investigation because the readers can be interesting to know a good suppler.
Reply: Agreed and revised.
1) Lines 104-106: “The chemical composition of the SHTB, in accordance with the quality specification from manufacturer (Eagle Metalware co., ltd.), is presented in Table 1.”
2) Lines 120-121: “Figure 2(a) presents the vibration test machine (RFDA HT650, IMCE, Belguum), which was used to measure Young’s modulus.”
3) Lines 145-147: “Figure 3. As shown in Figure 4, two clamps were utilized to secure the test specimen to the testing machine, which is the universal testing machine CMT5305 of MTS Indus-trial Systems (China) Co., Ltd., Shenzhen, China.”
Comment 2: An explain about the various colour found the samples should be discussed in the manuscript because this is curious results, this can be due to the different oxidation grade.
Reply: Thanks for your constructive advice. The authors checked the references and found that the oxidation is the main reason to change the metal’s surface color, but it is dependent on the oxidation products under high temperatures. It has been added into the revised manuscript, as follows:
1) Lines 163-166: “Studies [57,58] have shown that oxidation on the bolt surface is the main reason for its color change. At the same time, due to the difference in oxidation products when oxidation occurs in different temperature ranges, the bolts will show different colors after experiencing different temperatures.”
Comment 3: My specific comments are the following;
To add reference in line 160 of page 5 "istics [REF]. "
To insert reference in line 283 of page 10 "well predict the ultimate strain reduction factor of SHTB. Thus, the authors [REF] recommended"
To include reference in line 300 of the page 11 "higher than 600℃, as shown in Figure 13(b). Therefore, the authors [REF] recommended predic-"
Reply: The authors agree with this comment, and the discussion has been improved in the revised manuscript. For the second and third comments, the formulas are proposed in this study, not come from the published paper. To avoid misunderstanding, the authors also revised them. The changes are as follows:
1) Lines 168-170: “Visible necking phenomenon and shear lips can be seen from these photos, which be-comes more significant with the increasing temperature, also indicating that SHTB possesses ductile failure characteristics [2,13].”
2) Lines 293-295: “Thus, it is recommended to use Eq. (7) as the prediction formula for the ultimate strain reduction factor of SHTB. This formula is in good agreement with the experimental data.”
3) Lines 310-312: “Therefore, it is recommended to use Eq. (8) as the prediction formula for the reduction factor of percentage reduction of area of SHTB. This formula is in good agreement with the experimental data.”
Reviewer 2 Report
Comments and Suggestions for Authors
Comments and Suggestions for Authors
In this paper, the authors designed and conducted experiments on SHTBs under elevated temperatures, including both vibration and tensile coupon tests. Based on the test data, the stress-strain curves and key mechanical properties such as Young’s modulus, yield stress, ultimate stress, ultimate strain, percentage elongation, cross-sectional area reduction, and failure strain were obtained and analyzed for various high-temperature conditions.
The document is interesting and well-structured showing a full description of mechanical properties characterization under different temperatures. Some comments and suggestions for authors could be considered.
1. Include the manufacturer's commercial name for Grade 14.9 SHTB.
2. Indicate the manufacturer's name, city and country of origin of all experimental equipment used in the study i.e. (name, city, country).
3. In this study the authors assumed Poisson’s ratio as 0.3 for Grade 14.9 SHTB. However, it is well known the temperature variability of Poisson’s ratio. Can the authors explain how this variability of the Poisson’s ratio can affect the Young´s modulus model used in this study?
4.The three-stage suggested models describe the full uniaxial stress-strain curves of SHTB material under various elevated temperatures. Can this model be adapted to be used to predict different SHTB materials?
Author Response
In this paper, the authors designed and conducted experiments on SHTBs under elevated temperatures, including both vibration and tensile coupon tests. Based on the test data, the stress-strain curves and key mechanical properties such as Young’s modulus, yield stress, ultimate stress, ultimate strain, percentage elongation, cross-sectional area reduction, and failure strain were obtained and analyzed for various high-temperature conditions.
The document is interesting and well-structured showing a full description of mechanical properties characterization under different temperatures. Some comments and suggestions for authors could be considered.
Reply: Thanks for your positive comment.
Comment 1: Include the manufacturer's commercial name for Grade 14.9 SHTB..
Reply: Agreed and revised.
1) Lines 104-106: “The chemical composition of the SHTB, in accordance with the quality specification from manufacturer (Eagle Metalware co., ltd.), is presented in Table 1.”
Comment 2: Indicate the manufacturer's name, city and country of origin of all experimental equipment used in the study i.e. (name, city, country).
Reply: The authors do not agree with this comment.
1) Lines 120-121: “Figure 2(a) presents the vibration test machine (RFDA HT650, IMCE, Belguum), which was used to measure Young’s modulus.”
2) Lines 145-147: “Figure 3. As shown in Figure 4, two clamps were utilized to secure the test specimen to the testing machine, which is the universal testing machine CMT5305 of MTS Indus-trial Systems (China) Co., Ltd., Shenzhen, China.”
Comment 3: In this study the authors assumed Poisson’s ratio as 0.3 for Grade 14.9 SHTB. However, it is well known the temperature variability of Poisson’s ratio. Can the authors explain how this variability of the Poisson’s ratio can affect the Young´s modulus model used in this study?
Reply: Thanks for your question.
Jean et al. [1] pointed out that for the cold-formed steel under a temperature range 20 to 700 , the Poisson ratio will increase with the increasing of temperature, and its value ranges from 0.28 to 0.31. Even the material of SHTB is different from the cold-formed steel, but the data can be used as a reference.
ASTM E1876-15 suggested to calculated the Young’s modulus by using Eqs. (1) and (2) (see the manuscript). It can be found that the Poisson ratio can change the value of , and introduce to an error of Young’s modulus.
It is easy to find that the value of decreases with the increase of Poisson ratio. However, for a Poisson ratio ranges from 0.28 to .32, the ratio of (v) and (0.3) will range from 1.005 to 0.995. In other word, replacing (v) by (0.3), the introduced error is smaller than 0.5%, the accuracy meets engineering application requirements. Therefore, the effect of Poisson's ratio on Young's modulus model in this study can be ignored.
[1] Jean C Batista Abreu, Luiz M C Vieira, Metwally H Abu-Hamd & Benjamin W Schafer. Review: development of performance-based fire design for cold-formed steel. Fire Science Reviews, 2014, Volume 3, Article number 1.
Reviewer 3 Report
Comments and Suggestions for Authors In their work, the authors of the publication raised an important issue. The reliability of building structures under extreme conditions is critical. Knowledge and understanding of the behaviour of building structures in conditions of elevated temperatures is extremely important. Unfortunately, it is still impossible to exclude the risks of fires arising from improper operation of equipment, storage, use of incorrect building materials and cladding, and terrorist acts.
Various types of connections such as welding and bolt connections are widely used in construction. Each of them has its pros and cons. On the one hand, welding allows to obtain an integral connection, with the possibility of obtaining high properties of the material in the welding zone. On the other hand, not all metallic materials have good weldability. The higher the alloying content, the more difficult it is to weld the material. There is also the problem of thermal effects in the weld zone. Thus, none of the methods can be excluded from application. Bolt connections avoid the influence of the heat affected zone created by welding on the characteristics of the connection, but connections of equal strength may require a large number of bolts, especially for high-strength steel structures.
Bolt connection locations are hazardous points in the structure with an increased risk of failure. Therefore, the authors point out that using a 14.9 grade extra high strength bolt (SHTB) can significantly reduce the number of bolts, which will provide material and weight savings to the structure. However, the authors of the paper pointed out that despite a significant number of studies where the behaviour of different structural steels at elevated temperatures has been investigated, due to differences in material grade and manufacture, the mechanical properties of steel bolt are different from structural steel. Existing studies specifically on bolts have shown that the mechanical properties of the products decrease significantly in the temperature range of 300-400 °C. However, these studies have focused on conventional high-strength bolts such as Grade 8.8 and 10.9. SHTB, which has a higher ultimate stress than its conventional counterparts, has not been the subject of similar high-temperature mechanical properties studies.
The authors pointed out such a research gap. And also the lack of data on SHTB in current fire resistance design guidelines including BS EN 1993-1-2 [49], BS 5950-9 [50] and GB 51249-2017 [51], which limits the application of SHTB in fire resistance design practice. The aim of this work was to fill the gap on the investigation of high temperature mechanical properties of SHTB.
To fill this gap, the authors designed and conducted an experimental study of the mechanical properties and stress-strain relationship of SHTB at elevated temperatures. Young's modulus, yield strength, ultimate strength, ultimate strain, percentage elongation, cross-sectional area reduction, and fracture strain of SHTB at different temperatures were obtained. The variation of mechanical properties with increasing temperature was then quantitatively analysed and compared with publicly available data of different types of high-strength bolts. Finally, a new three-stage model was developed to describe the stress-strain relationship of SHTB under fire conditions.
The aim of the work was achieved by the authors. I would particularly like to highlight how detailed and excitingly the authors have demonstrated that modelling the behaviour of materials and products made from them based on data on other materials, i.e. economising on experimental studies and using broad assumptions, is a flawed approach. This is especially critical in the construction industry, where errors such as underestimating the behaviour of products under extreme conditions can lead to the loss of many lives.
The introductory part of article is quite detailed. The methodology of experimental studies, further analysis steps and modelling is described in details.
The figures presented are appropriate and easy to interpret.
References are appropriate. Recommendation for the authors is to add to the list of references, if possible, more articles for the last 5 years, revealing the current state of the art of the issue.
Author Response
In their work, the authors of the publication raised an important issue. The reliability of building structures under extreme conditions is critical. Knowledge and understanding of the behaviour of building structures in conditions of elevated temperatures is extremely important. Unfortunately, it is still impossible to exclude the risks of fires arising from improper operation of equipment, storage, use of incorrect building materials and cladding, and terrorist acts.
Various types of connections such as welding and bolt connections are widely used in construction. Each of them has its pros and cons. On the one hand, welding allows to obtain an integral connection, with the possibility of obtaining high properties of the material in the welding zone. On the other hand, not all metallic materials have good weldability. The higher the alloying content, the more difficult it is to weld the material. There is also the problem of thermal effects in the weld zone. Thus, none of the methods can be excluded from application. Bolt connections avoid the influence of the heat affected zone created by welding on the characteristics of the connection, but connections of equal strength may require a large number of bolts, especially for high-strength steel structures.
Bolt connection locations are hazardous points in the structure with an increased risk of failure. Therefore, the authors point out that using a 14.9 grade extra high strength bolt (SHTB) can significantly reduce the number of bolts, which will provide material and weight savings to the structure. However, the authors of the paper pointed out that despite a significant number of studies where the behaviour of different structural steels at elevated temperatures has been investigated, due to differences in material grade and manufacture, the mechanical properties of steel bolt are different from structural steel. Existing studies specifically on bolts have shown that the mechanical properties of the products decrease significantly in the temperature range of 300-400 °C. However, these studies have focused on conventional high-strength bolts such as Grade 8.8 and 10.9. SHTB, which has a higher ultimate stress than its conventional counterparts, has not been the subject of similar high-temperature mechanical properties studies.
The authors pointed out such a research gap. And also the lack of data on SHTB in current fire resistance design guidelines including BS EN 1993-1-2 [49], BS 5950-9 [50] and GB 51249-2017 [51], which limits the application of SHTB in fire resistance design practice. The aim of this work was to fill the gap on the investigation of high temperature mechanical properties of SHTB.
To fill this gap, the authors designed and conducted an experimental study of the mechanical properties and stress-strain relationship of SHTB at elevated temperatures. Young's modulus, yield strength, ultimate strength, ultimate strain, percentage elongation, cross-sectional area reduction, and fracture strain of SHTB at different temperatures were obtained. The variation of mechanical properties with increasing temperature was then quantitatively analysed and compared with publicly available data of different types of high-strength bolts. Finally, a new three-stage model was developed to describe the stress-strain relationship of SHTB under fire conditions.
The aim of the work was achieved by the authors. I would particularly like to highlight how detailed and excitingly the authors have demonstrated that modelling the behaviour of materials and products made from them based on data on other materials, i.e. economising on experimental studies and using broad assumptions, is a flawed approach. This is especially critical in the construction industry, where errors such as underestimating the behaviour of products under extreme conditions can lead to the loss of many lives.
The introductory part of article is quite detailed. The methodology of experimental studies, further analysis steps and modelling is described in details.
The figures presented are appropriate and easy to interpret.
Reply: Thanks for your positive comment.
Comment 1: References are appropriate. Recommendation for the authors is to add to the list of references, if possible, more articles for the last 5 years, revealing the current state of the art of the issue.
Reply: The authors agree with this comment and have added some new references in the same area. The revised manuscript as follows:
1) The added references:
- H. Gao, B. Yang, Z.K. Liu, M. Elchalakani, and L. Shen. Study on Post-Fire Constitutive Model and Fracture Performance of Grade 14.9 Superhigh Tension Bolts. Journal of Structural Engineering, 2025. (Accepted)
- Huseyin Saglik, Airong Chen, and Rujin Ma. Ductile fracture of high-strength bolts under combined actions at elevated tem-peratures. Journal of Constructional Steel Research, 2024, Volume 213, pp. 108437.
2) Lines 43-44: “SHTBs are now commercially available, enhancing the options for robust structural connections in key markets such as Japan [8, 9], the United States [10-12], and China [13,14].”
3) Lines 82-85: “Huseyin et al. [50] presented a study on the ductile fracture of Grade 8.8 or 10.9 bolts under fire conditions. Based on the test results from tension, shear, and combine tension-shear tests, they calibrated the fracture parameters and investigated the influence of displacement at fracture values on the results by a three-bolted inclined shear test.”